# Polyacrylic-Co-Maleic-Acid-Coated Magnetite Nanoparticles for Enhanced Removal of Heavy Metals from Aqueous Solutions

Rawan Mlih [1,2,*], Jonathan Suazo-Hernández [3], Yan Liang [4], Etelka Tombácz [5], Roland Bol [1,6]
and Erwin Klumpp [1,*]

1 Institute of Bio- and Geosciences, Agrosphere (IBG–3), Forschungszentrum Juelich (FZJ), 52425 Juelich, Germany
2 Institute for Environmental Research, Biology 5, RWTH Aachen University, 52074 Aachen, Germany
3 Center of Plant, Soil Interaction and Natural Resources Biotechnology, Scientific and Biotechnological Bioresource Nucleus, Universidad de La Frontera, Temuco 4780000, Chile
4 School of Resources, Environment and Materials, Guangxi University, Nanning 530004, China
5 Soós Ernő Research and Development Center, University of Pannonia, H-8800 Nagykanizsa, Hungary
6 School of Natural Sciences, Environment Centre Wales, Bangor University, Bangor LL57 2UW, Gwynedd, UK
* Correspondence: r.mlih@fz-juelich.de (R.M.); e.klumpp@fz-juelich.de (E.K.)

**Abstract:** The physicochemical properties of ligand-coated nanoparticles make them superior adsorbents for heavy metals from water. In this study, we investigate the adsorption potential of novel polyacrylic-co-maleic-acid-coated magnetite nanoparticles (PAM@MNP) to remove $Pb^{2+}$ and $Cu^{2+}$ from an aqueous solution. We argue that modifying the surface of MNP with PAM enhances the physicochemical stability of MNP, improving its ability to remove heavy metals. The adsorption kinetics data show that PAM@MNP attained sorption equilibrium for $Pb^{2+}$ and $Cu^{2+}$ after 60 min. The kinetics data are fitted accurately by the pseudo-first-order kinetic model. The calculated Langmuir adsorption capacities are 518.68 mg g$^{-1}$ and 179.81 mg g$^{-1}$ for $Pb^{2+}$ and $Cu^{2+}$, respectively (2.50 mmol g$^{-1}$ and 2.82 mmol g$^{-1}$ for $Pb^{2+}$ and $Cu^{2+}$, respectively). The results indicate that PAM@MNP is a very attractive adsorbent for heavy metals and can be applied in water remediation technologies.

**Keywords:** coated magnetite nanoparticles; $Pb^{2+}$; $Cu^{2+}$; adsorption kinetics; adsorption isotherms





## 1. Introduction

In recent years, nanomaterials have gained a lot of attention as adsorbents for heavy metals due to their high adsorption ability and reusability. The physicochemical properties of magnetite nanoparticles (MNP) such as surface area, particle size, and surface charge offer great applicability for the removal of heavy metals from water [1]. MNP used in bare or modified form exhibit high efficiency for removing Pb, Cu, Zn, Mn, Hg, and Cr from aqueous systems [2–4]. The adsorption of heavy metals to MNP benefits from the existence of oxygen-containing functional groups on the magnetite surface, which form complexes with heavy metal ions [5]. MNP are considered eco-friendly materials, affordable, easy to use, and susceptible to separation from water solution by magnetic separation [6].

Bare MNP have a great tendency to aggregate or be oxidized in an aqueous solution, which is why MNP are coated with a wide range of organic and inorganic materials [7]. Coatings provide steric stability for the nanoparticles, diminish their aggregation, and alter the surface charge, which in turn influences the electrostatic interaction between the nanoparticles and the sorptive [8].

Polyelectrolyte-coated MNP with excess ligands have a higher potential for $Pb^{2+}$ and $Cu^{2+}$ removal than bare MNP [9]. Different studies have demonstrated that coatings containing organic polymers such as poly(acrylic acid) (PAA), humic acid (HA), or carboxylic methyl cellulose (CMC) enhance the adsorption of heavy metals (i.e., $Pb^{2+}$ and $Cu^{2+}$) due to their possession of reactive ligands, i.e., carboxyl groups (−COOH), which play a prominent role in the adsorption process through complex formation and ion exchange [10–15].

To the best of our knowledge, poly(acrylic acid-co-maleic acid) (PAM) with respect to its metal-ion-binding ability has been rarely addressed in the literature. Existing studies focus on using PAM in a hybrid ultrafiltration–electrolytic process to remove $Cu^{2+}$ and $Pb^{2+}$ from water [16,17]. The high retention of heavy metals observed in these studies has been attributed to the strong ligand interaction between metal ions and the polymer, favored by the presence of carboxyl groups in the polymer structure.

PAM coating was used for the first time on MNP to improve the colloidal stability of MNP for biomedical applications [18]. In comparison to other coatings such as PAA, PAM revealed excellent abilities to be affixed to MNP because maleic acid forms metal−carboxylate complexation at oxide/electrolyte interfaces [19]. Jang et al. [20] showed that each poly(acrylic acid-co-maleic acid) ($M_w$ = ~3000 amu) used as a coating for nanoparticles has an equal number of acrylic acid and maleic acid monomers that possess an ample number of carboxyl groups (−COOH). However, some of these groups can be conjugated to the nanoparticle surface, while others remain free. Additionally, the geometric matching between the carboxylate groups of maleic acid moieties and the surface sites of the crystalline phase of magnetite supports inner-sphere complex formation. PAM exhibited high adsorption affinity to MNP and high dilution resistance associated with the very low concentration of free PAM in solution compared to PAA [18]. This physicochemical stability of PAM@MNP makes the material a candidate for environmental remediation from heavy metals. Based on this, the present study investigates for the first time $Pb^{2+}$ and $Cu^{2+}$ removal from an aqueous solution using PAM@MNP by performing adsorption kinetic and isotherm experiments and modeling.

## 2. Materials and Methods

### 2.1. Materials

The synthesis and coating of PAM@MNPs were achieved at the Department of Physical Chemistry and Materials Science, University of Szeged in Hungary. The preparation and coating details are described elsewhere [8,18,21]. FTIR-ATR spectra for PAM@MNP are illustrated in a previous study [22], in addition, TEM images for MNP and PAM@MNP are shown in Figure S1. Briefly, magnetite ($Fe_3O_4$) nanoparticles were prepared by co-precipitation of $Fe^{2+}$ and $Fe^{3+}$ salts in an alkaline (NaOH) medium and purified by dialysis and magnetic separation. For coating, poly(acrylic acid-co-maleic) acid (PAM) $M_w$ 3000 Da; 15.9 mmol COOH/g polymer was added to magnetite to load 1.3 mmol PAM/g magnetite. The time of adsorption was set to 1 h and pH was adjusted to ~6.5 using NaOH solution. Then, 10 g PAM-coated magnetite was dispersed in 200 mL ultrapure water. The produced PAM@MNP was negatively charged in the pH range from 3 to 10 (Figure S2) [18].

Stock solutions of heavy metals were prepared from copper chloride dihydrate ($CuCl_2 \cdot 2H_2O$), (purity ≥ 99.0%), and lead chloride ($PbCl_2$) salts (purity ≥ 98.0%), (Sigma Aldrich, Germany) by dissolving the proper amounts of the salt in Milli-Q water. The lead salt was stirred for 2 h at 50 °C using a magnetic stirrer to attain complete dissolution. The solutions were further diluted to the desired concentrations for the experiments.

### 2.2. Instruments and Analytical Methods

The surface area of PAM@MNP measured by BET was 75.9 $m^2\ g^{-1}$. It was measured by nitrogen adsorption and desorption at 77 K using isotherms 5-point BET, Autosorb-1 analyzer (Quantachrome, Syosset, NY, USA). The samples were degassed at 200 °C under helium flow for 1 h before analysis. The zeta potential and hydrodynamic diameter of PAM@MNP under pH 8.5 and ionic strength of 1 mM measured using Zetasizer (Malvern Instruments GmbH, Herrenberg, Germany) were −62 ± 3.4 mV and 126 ± 1.5 nm, respectively. A high-speed centrifuge (Avanti JXN-30, Beckman Coulter, Ypasadena, CA, USA) was used to separate the nanoparticles from the solution. A digital pH meter (Metrohm, Herissau, Switzerland) was used for pH adjustment of the solutions. The concentrations of metal ions were measured using ICP-OES (Thermo Scientific, Karlsruhe, Germany).

### 2.3. Batch Adsorption Experiments

The surface area adsorption isotherms were determined by adding 0.2 g L$^{-1}$ PAM@MNP suspension into 100 mL vials containing 50 mL of Pb$^{2+}$ and Cu$^{2+}$ metal solution with different concentrations. The pH values for all experiments were adjusted using hydrochloric acid (HCl) and sodium hydroxide (NaOH) of 0.01 M. The effect of pH on the adsorption was studied in the range from 3 to 9. The metal concentration was set to 10 mg L$^{-1}$. The adsorption kinetics were measured up to 480 min at 20 °C using 50 mL Pb$^{2+}$ or Cu$^{2+}$ solution (10 mg L$^{-1}$) mixed with 0.2 g L$^{-1}$ sorbent. PAM@MNP was mixed with the metal solution in a horizontal shaker at 180 rpm. After adsorption experiments, the liquid phase was separated by centrifuging at 48,000 rpm for 30 min (Avanti JXN-30, USA) and then filtered using 0.1 μm syringe filter (Sartorius, Göttingen, Germany).

The amount of metal adsorbed on PAM@MNP was calculated based on the difference in metal concentrations in the aqueous solution before and after the adsorption experiment according to Equation (1)

$$q_e = \frac{(C_0 - C_e) \times V}{m} \tag{1}$$

where $C_0$ and $C_e$ are the initial and equilibrium concentrations of Pb$^{2+}$ and Cu$^{2+}$ (mg L$^{-1}$ or mmol L$^{-1}$), $q_e$ represents the adsorbed amount (mg g$^{-1}$ or mmol g$^{-1}$), $m$ denotes the mass of PAM@MNP (g), and $V$ is the volume of the solution (L). The removal efficiency R% was obtained from Equation (2)

$$R\% = \frac{(C_0 - C_e)}{C_0} \times 100\% \tag{2}$$

where $C_0$ and $C_e$ (mg L$^{-1}$) are the same as in Equation (1).

### 2.4. Kinetics Models of Adsorption

In order to investigate the rate and mechanism of Pb$^{2+}$ and Cu$^{2+}$ adsorption, pseudo-first-order and pseudo-second-order equations were applied to fit the data according to Equations (3) and (4)

$$q_t = q_e \left(1 - e^{-k_1 t}\right) \tag{3}$$

$$q_t = \frac{q_e^2 k_2 t}{q_e k_2 t + 1} \tag{4}$$

where $q_e$ and $q_t$ are the amounts of heavy metals adsorbed (mg g$^{-1}$ or mmol g$^{-1}$) at time of equilibrium and at time $t$ (min), respectively, and $k_1$ (min$^{-1}$) and $k_2$ (g mg$^{-1}$ min$^{-1}$ or g mmol$^{-1}$ min$^{-1}$) are the rate constants of the pseudo-first-order and second-order model of adsorption, respectively. The initial adsorption rate $h$ (mg g$^{-1}$ min$^{-1}$ or mmol g$^{-1}$ min$^{-1}$) of the second-order model was obtained from $k_2 q_t^2$.

### 2.5. Adsorption Isotherms

The experimental data for Pb$^{2+}$ and Cu$^{2+}$ adsorption onto PAM@MNP were fitted to Langmuir and Freundlich isotherm models as given in Equations (5) and (6), respectively,

$$q_e = \frac{q_m K_L C_e}{1 + K_L C_e} \tag{5}$$

where $q_e$ is the amount of metal adsorbed per gram of the adsorbent at equilibrium (mg g$^{-1}$ or mmol g$^{-1}$), $q_m$ is the Langmuir adsorption capacity (mg g$^{-1}$ or mmol g$^{-1}$), $K_L$ is Langmuir isotherm constant (L mg$^{-1}$ or L mmol$^{-1}$), and $C_e$ is the equilibrium concentration of adsorbate (mg L$^{-1}$ or mmol L$^{-1}$).

$$q_e = K_F C_e^{1/n} \tag{6}$$

where $q_e$ refers to the amount of metal adsorbed (mg g$^{-1}$ or mmol g$^{-1}$) at equilibrium, $C_e$ is the equilibrium concentration of the metal (mg L$^{-1}$ or mmol L$^{-1}$), $K_F$ is the Freundlich constant (mg g$^{-1}$ or mmol g$^{-1}$), and *1/n* refers to the linearity of adsorption.

## 3. Results and Discussion

### 3.1. Effect of pH

Solution pH is an important parameter in the adsorption process of heavy metals as it affects the speciation of the metal and the surface charge of the adsorbent and its functional groups [1]. Figure 1 shows that at low pH, the removal efficiency of Pb$^{2+}$ and Cu$^{2+}$ by PAM@MNP decreased. This can be attributed to a high concentration of H$^{+}$ ions that compete with heavy metal ions on the adsorbent surface [23]. As the pH value changed from 3 to 6, the removal efficiency increased from 28.2% to 95.7% for Pb$^{2+}$ and from 3.3% to 97.4% for Cu$^{2+}$. The high removal at elevated pH can be attributed to possible carboxyl group dissociation of PAM, which facilitates the electrostatic attraction between the positively charged metals and $-$COO$^{-}$ [24,25]. The dissociation of carboxyl groups is confirmed by the *pKa* values of maleic acid at 1.83 [26] and the *pKa* of acrylic acid at 4.25 [27]. Guan at al. [28] found that the high removal efficiency of Pb$^{2+}$ using PAA-grafted MNP at high pH is caused by the dissociation of carboxyl groups of PAA, which facilitates the binding of Pb$^{2+}$ ions. Paulino et al. [11] showed that, at pH 5.5, Pb$^{2+}$ and Cu$^{2+}$ ions may form different types of complexes with carboxylic groups on PAA-coated chitosan-based hydrogels with a magnetite core.

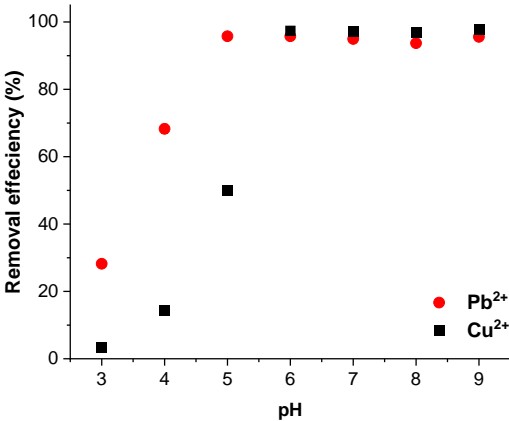

**Figure 1.** Effect of pH on the removal efficiency of Pb$^{2+}$ and Cu$^{2+}$ by PAM@MNP. Adsorbent dose: 0.2 g L$^{-1}$; initial Cu$^{2+}$ and Pb$^{2+}$ concentration: 10 mg L$^{-1}$; temperature: 20 °C; contact time: 24 h.

However, under alkaline conditions, possible precipitation of cations as hydroxides may take place in the solution or on the nanoparticle surfaces. Therefore, adsorption cannot be considered the only mechanism for the removal of heavy metals. Several studies have demonstrated that at pH > 6, Pb$^{2+}$ removal occurs via precipitation as lead hydroxide [29–31]. Cu$^{2+}$ exists as free ions in the water at pH < 6. Its precipitation as copper hydroxide occurs at pH 6.5 [32,33]. Based on this, the pH was set to 6 for isotherm and kinetic studies to avoid the effect of precipitation.

### 3.2. Metals Adsorption Kinetics and Isotherms

The kinetics experiments were carried out at pH 6. The experimental results shown in Figure 2 suggested that the sorption equilibrium was achieved after 60 min for both metals with a removal efficiency >90% (not shown here). The adsorption rate was predicted by applying the kinetics models. The adsorption kinetic data could be fitted better to the pseudo-first-order kinetic model, as shown in Figure 2 and Table 1 (R$^2$ = 0.95 and 0.99 for Pb$^{2+}$ and Cu$^{2+}$, respectively). This indicates that the mechanism of adsorption at the applied concentration of heavy metals might have been driven by electrostatic attraction forces between the positively charged metal ions and the negatively charged PAM@MNP [34].

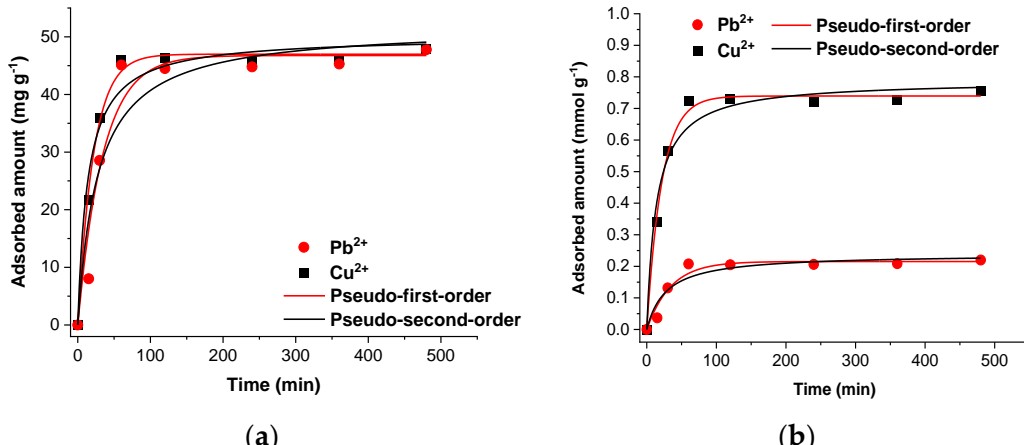

**Figure 2.** Adsorption kinetics of $Pb^{2+}$ and $Cu^{2+}$ (**a**) in mg $g^{-1}$ and (**b**) in mmol $g^{-1}$ onto PAM@MNP fitted to pseudo-first-order and pseudo-second-order models. Adsorbent concentration: 0.2 g $L^{-1}$; initial metal concentration: 10 mg $L^{-1}$; pH: 6; temperature: 20 °C.

**Table 1.** Parameters of pseudo-first-order and pseudo-second-order kinetic models for $Pb^{2+}$ and $Cu^{2+}$ adsorption onto PAM@MNP.

| Kinetic Model | Parameter | Metal | |
|---|---|---|---|
| | | $Pb^{2+}$ | $Cu^{2+}$ |
| Pseudo-first-order | $q_e$ (mg $g^{-1}$); (mmol $g^{-1}$) | 46.76; 0.22 | 46.98; 0.73 |
| | $k_1$ ($min^{-1}$) | 0.03 | 0.05 |
| | $R^2$ | 0.95 | 0.99 |
| Pseudo-second-order | $q_e$ (mg $g^{-1}$); (mmol $g^{-1}$) | 51.89; 0.23 | 50.11; 0.79 |
| | $k_2$ (g $mg^{-1}$ $min^{-1}$); (g $mmol^{-1}$ $min^{-1}$) | $6.99 \times 10^{-4}$; 0.15 | $1.45 \times 10^{-3}$; 0.09 |
| | h (mg $g^{-1}$ $min^{-1}$); (mmol $g^{-1}$ $min^{-1}$) | 1.9; $8.68 \times 10^{-3}$ | 3.6; 0.06 |
| | $R^2$ | 0.91 | 0.97 |

The adsorption isotherms of $Pb^{2+}$ and $Cu^{2+}$ onto PAM@MNP are shown in Figure 3. The Langmuir model shows better fitting for the isotherms compared to the Freundlich model ($R^2$ = 0.92 for both metals; Table 2). The Langmuir adsorption capacity ($q_m$, monolayer adsorption coverage) for $Pb^{2+}$ and $Cu^{2+}$ is 518.68 mg $g^{-1}$ and 179.81 mg $g^{-1}$, respectively, which corresponds to 2.50 mmol $g^{-1}$ for $Pb^{2+}$ and 2.82 mmol $g^{-1}$ for $Cu^{2+}$ (Figure 3b). The results indicate that the surface of the adsorbent is homogenous in nature, and all binding sites were uniformly occupied by metal ions until a monolayer of heavy metal ions developed on the surface of the adsorbent [35]. $Pb^{2+}$ and $Cu^{2+}$ ions were almost completely removed (>96%) for initial concentrations up to 70 mg $L^{-1}$ and 30 mg $L^{-1}$, respectively (not shown here). The reason behind a lower initial concentration for the $Cu^{2+}$ metal was to avoid the effect of precipitation, which is enhanced at higher initial concentrations of the metal [36]. However, the adsorption results in mmol $g^{-1}$ are higher for $Cu^{2+}$ than for $Pb^{2+}$ which indicates that the precipitation of copper hydroxide on the surface might still be possible even at low concentrations at the applied pH value of 6 [37]. The high adsorbed amount for both metals can be attributed to the different forms of carboxyl groups such as −COOH and −COO$^-$ contained in PAM@MNP at pH 6 [18]. We assume that these groups are involved in the formation of surface complexes with $Pb^{2+}$ and $Cu^{2+}$ [16,38,39]. The role of −COOH groups in the adsorption process has been reported for PAA-coated magnetite used as an adsorbent for $Pb^{2+}$ [28,39] and $Cu^{2+}$ ions [11,40]. Protons on −COOH groups can serve as exchangeable ions [38] and upon dissociation, the −COO$^-$ groups provide binding sites for heavy metals cations.

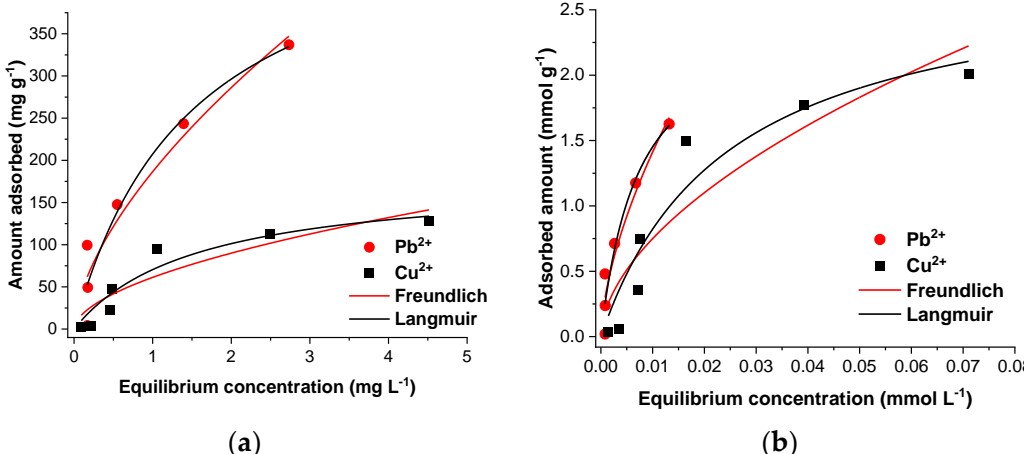

**Figure 3.** Langmuir and Freundlich adsorption isotherm models (**a**) in mg g$^{-1}$ and (**b**) in mmol g$^{-1}$ for the adsorption of Pb$^{2+}$ and Cu$^{2+}$ onto PAM@MNP. Symbols represent the experimental data. Initial concentration: 0.5–70 mg L$^{-1}$; adsorbent concentration: 0.2 g L$^{-1}$; pH: 6; temperature: 20 °C; contact time: 24 h.

**Table 2.** Langmuir and Freundlich isotherm parameters for the adsorption of Pb$^{2+}$ and Cu$^{2+}$ onto PAM@MNP.

| Isotherm | Parameter | Metal | |
|---|---|---|---|
| | | Pb$^{2+}$ | Cu$^{2+}$ |
| Langmuir | $q_m$ (mg g$^{-1}$); (mmol g$^{-1}$) | 518.68; 2.50 | 179.81; 2.82 |
| | $K_L$ (L mg$^{-1}$); (L mmol$^{-1}$) | 0.67; 138.06 | 0.64; 40.99 |
| | R$^2$ | 0.92 | 0.92 |
| Freundlich | $K_F$ (mg g$^{-1}$); (mmol g$^{-1}$) | 186.54; 23.18 | 61.27; 9.63 |
| | n | 1.62; 1.62 | 1.80; 1.80 |
| | R$^2$ | 0.91 | 0.86 |

The calculated Langmuir adsorption capacity of PAM@MNP for Pb$^{2+}$ and Cu$^{2+}$ metals is much higher than some reported values for MNP coated with comparable coating materials. As the amount of PAM coating was about 1 mmol/g, given in −COOH equivalent [18], the 2–3 times higher amounts of Me$^{2+}$ adsorbed indicate surface precipitation or adsorption on MNP.

## 4. Conclusions

In this study, we investigated PAM@MNP as a new adsorbent for the high removal efficiency of heavy metals from contaminated water. The achieved high adsorption of Pb$^{2+}$ and Cu$^{2+}$ on PAM@MNP was attributed to a PAM coating. In comparison to the frequently used PAA coating, PAM affixes better to MNP by forming metal−carboxylate complexes at oxide/electrolyte interfaces, which enhances the physicochemical stability and regeneration potential of MNP. The adsorption of Pb$^{2+}$ and Cu$^{2+}$ onto PAM@MNP was fast and efficient. PAM@MNP showed >90% removal efficiency in 60 min. and Langmuir adsorption capacities of 518.68 mg g$^{-1}$ and 179.81 mg g$^{-1}$ for Pb$^{2+}$ and Cu$^{2+}$, respectively. The findings provide evidence of the suitability of the PAM@MNP application for the removal of Pb$^{2+}$ and Cu$^{2+}$ from water. However, PAM@MNP regeneration and separation from the aqueous solution still need to be elucidated in further steps.

**Supplementary Materials:** The following supporting information can be downloaded at: https://www.mdpi.com/article/10.3390/colloids7010005/s1, Figure S1: TEM images for (a) bare MNP and (b) PAM-coated MNPs; Figure S2: pH-dependent surface charging of bare and PAM-coated MNPs.

**Author Contributions:** Conceptualization: R.M. and E.K.; methodology: R.M., E.T. and Y.L.; software: J.S.-H. and R.M.; validation: E.K. and E.T.; formal analysis: J.S.-H. and R.M.; investigation: R.M.; writing—original draft preparation: R.M.; writing—review and editing: all authors; funding acquisition: R.B.; resources: E.K. and E.T.; supervision: E.K. and R.B.; project administration: R.B. All authors have read and agreed to the published version of the manuscript.

**Funding:** This research was funded by the German Federal Ministry of Education and Research (BMBF) under the Palestinian German Science Bridge (PGSB) program grant number 01DH16027.

**Data Availability Statement:** Not applicable.

**Acknowledgments:** Yan Liang acknowledges a scholarship from the China Scholarship Council. The authors thank Volker Nischwitz for support during the analyses at the Central Institute for Engineering, Electronics and Analytics (ZEA-3) at FZJ.

**Conflicts of Interest:** The authors declare no conflict of interest.

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
