# Peer review of "Polyacrylic-Co-Maleic-Acid-Coated Magnetite Nanoparticles for Enhanced Removal of Heavy Metals from Aqueous Solutions"

_colloids, doi:10.3390/colloids7010005_

Round 1

Reviewer 1 Report

1. Please state the purity of used materials.

2. Line 91 - use the proper Celsius symbol and remove the space between the value and symbol according to SI units.

3. English needs to be checked by a native speaker as there are some mistakes

4. Line 130 adds an upper index for Pb2+ and Cu2+

5. Authors should use a lot higher concentrations of used metals 30 and 70 mg/L-1 are low concentrations which are easy to remove. I'd suggest experimenting with 500 or even 1000 ppm of metal ions to really show the possibility of the used material.

6. Provide SEM-EDS microphotographs to show the structure of obtained material and EDS to show the monolayer adsorption 

7. FTIR analysis must be added to prove that  the process is physical and not of chemical nature

8. Author can cite the below literature to enhance the information about the sorption process:

https://doi.org/10.1016/j.colsurfa.2021.127735

Author Response

Dear Mme/Sir,

Best regards

Reviewer 2 Report

Presently, it is difficult to judge the work of the author due to the lack of the basic information of PAM coated MNP. Although the author has referred to relevant references, it is difficult or inconvenient to determine which content is directly relevant. I suggest that author put those information in the supplementary file. In addition, there is no discussion on the adsorption mechanism, such as why the adsorption  of Cu and Pb ions are different, and adsorption is mainly based on PAM or MNP ?

Author Response

Dear Mme/Sir

Best regards

Reviewer 3 Report

In the submitted manuscript, Mlih et al. investigated the efficiency of the removal of Pb2+ and Cu2+ using polyacrylic-co-maleic acid-coated Fe2O3 nanoparticles (PAM@MNP). Detailed insights into the adsorption mechanism were gained from the adsorption isotherms and adsorption kinetics study. Additionally, the efficiency of removal was investigated at different pH of water. The maximum adsorption was achieved under pH 6. The ions’ precipitation process at the solution and the surface of NPs via low soluble hydroxides was observed at pH under 8. At low pH, the removal efficiency was low.

The manuscript is well-written and contains important results which should contribute to the further development of PAM@MNP. Hence, it should be published in one of the ongoing issues.

However, before publishing, the authors have to answer some questions and accept minor suggestions.

For example:

Why did they use such a high rate of centrifugation? 46000rpm 30 min? ( page 3, line 113)

What is Freundlich constant Kf or kf? Which one?

Please correct Pb2+ and Cu2+ ( Page 3, line 130)?

In conclusion, the authors said: „In this study, we suggested PAM@MNP as a new adsorbent for high removal efficiency of heavy metals from contaminated water.“ I am wondering, is it true? They investigated the removal of only Pb2+ and Cu2+ ions, not all heavy metals. Additionally, they did not investigate removal from contaminated water but only in pure water at different pH after adding the mentioned ions. The real water sample is a much more complex mixture, and the other effects should be investigated.

Author Response

Dear Mme/Sir

Best regards

Rawan

Round 2

Reviewer 1 Report

Thank you for you responses.

Reviewer 2 Report

OK, I understand the situation of the authors. it can be accepted.